# Connectivity and Obstacle Avoidance Method for Formation Tracking with Uncertain Multiple Nonholonomic Mobile Robots with Unknown Faults

1st Yanbing Han
*College of science*
*Liaoning University of Technology*
Jinzhou, China
hanyb0615@163.com

2nd Kewen Li
*College of science*
*Liaoning University of Technology*
Jinzhou, China
likewen2018@163.com

3rd Yongming Li
*College of science*
*Liaoning University of Technology*
Jinzhou, China
l_y_m_2004@163.com

*Abstract*—This paper delves into the challenge of ensuring connectivity and steering clear of obstacles for multiple uncertain nonholonomic mobile robots that face limitations in communication and sensing distances under the influence of unknown actuator defects. It operates under the assumption that the nonlinear dynamics of these robots are completely unknown. The necessary relative angles to sustain connectivity and simultaneously avoid obstacles are derived. Furthermore, an innovative strategy for obstacle avoidance that also maintains connectivity is introduced. Then, using these desired relative angles and performance functions that maintain connectivity and avoid collisions, a leader-follower formation tracker is designed to achieve connectivity maintenance, collision avoidance, and obstacle avoidance among robots. Simultaneously, to compensate for the effects of intermittent unknown actuator faults and enable the robots to continue interacting with their leader and uninterruptedly track the leader's time-varying reference trajectory during actuator failures, an adaptive law with unknown parameters is further proposed for time-varying formations, ensuring that all signals are semi-globally uniformly ultimately bounded. Finally, the stability analysis is performed using the Lyapunov equation.

*Index Terms*—Uncertain multiple nonholonomic mobile robots, fault tolerant control, adaptive control, connectivity-maintaining obstacle avoidance, leader-follower formation

## I. INTRODUCTION

Mobile robots have a wide range of practical applications in rescue, agricultural and civil tasks, especially in potential applications such as intelligent storage, automated logistics, monitoring and intelligent transportation. [1] delves into the formation control problem of multiple mobile robots with incomplete constraints.In the actual control environment, network connectivity between multiple robots and collision avoidance problems are of great significance for completing tasks related to formation control [2].

Formation control methods can be mainly divided into the following categories, virtual structure method, behavior-based method, graph theory-based method, artificial potential field method, leader-follow method [3] [4] [5],The virtual

This work is supported by National Natural Science Foundation of China under Grant U22A2043

structure method is simple to implement, but it is difficult to use in scenarios where the shape of the formation needs to be changed frequently. In addition, due to the need for a centralized structure, the communication load of computing nodes is large, and single-point hardware problems are prone to occur [6]. Compared with the virtual structure method, the behavior-based method adopts a distributed architecture, so it has lower communication requirements. However, due to the stability and robustness analysis, how to ensure the convergence of the formation shape still needs to be studied. In addition, behavior-based approaches are difficult to guarantee stability due to the difficulty of deriving mathematical models.Compared with other methods, the graph theory method is more suitable for scenarios where the communication topology changes dynamically, but the agent can only communicate with nearby agents.The potential field method is easy to deal with obstacle avoidance problems and is applied in real time. However, it is difficult to design a suitable potential field function, and the results given by the potential field method are often locally optimal. This article does not use any potential field function, but only changes the angle to deal with the obstacle avoidance problem.The approach of leader-follower is mathematically comprehensible and diminishes both the communication load and computational strain, thanks to its decentralized architecture. This method is widely used in the formation control problem of multiple mobile robots. In early research on formation control of non-complete mobile robots, only the kinematics of the robots were considered, and these jobs required perfect velocity tracking assumptions [7], and the uncertainties inherent in the dynamics of mobile robots were not considered.In order to solve these problems, researchers have proposed control methods, such as synovial control, adaptive control, and intelligent control. However, these methods cannot be applied to connectivity maintenance and collision avoidance issues.

In order to carry out the master-slave control design of multiple nonholonomic mobile robots more practically, many methods are studied to solve the problems of connectivity maintenance and collision avoidance.For example, a vision

sensor-based approach that considers visible constraints [8], uses potential-like functions to maintain connections between robots and avoid collisions [9] [10] [11]. Since the potential function may fall into a local minima, [12] proposes a decentralized control method using the dipole navigation function. [13] proposes a method based on the rotation matrix and [14] proposing a unified error transformation method that can achieve connectivity preservation and collision avoidance without using the potential function.

In addition, as the robot components age, the actuator may fail. A single robot failure can affect the entire system, making the system unstable, which in turn makes each robot uncontrollable, not only unable to maintain a normal formation structure, but also causing robots to collide and unable to maintain normal connectivity. In the previous literature, there is not much consideration of unknown and different fault handling, and an adaptive compensation control method is designed to effectively solve the problem of unknown faults in actuators.In [15], an adaptive fuzzy fault-tolerant control method based on directed switching graph theory is proposed for formation tracking of uncertain nonlinear multi-agent systems, which can simultaneously respond to the influence of sensor faults on unknown nonlinearity.In order to adapt to an infinite number of uncertain faults in a finite time, a new adaptive method for uncertain nonlinear systems is proposed for the first time in [16], which combines finite time theory with command filtering technology to achieve rapid adaptation to infinite uncertain faults. But it doesn't apply to real-world engineering systems.

In contrast to previous studies in the field, this paper's primary contributions are outlined below.

- In comparison to other studies, the issue of localIn comparison to other studies, the issue of local crafting a controller that lack the issue of local minima is addressed by crafting a controller that lacks a potential function. Additionally, an adaptive law for parameters has been developed to tackle unknown actuator failures in the dynamic behavior of mobile robots.
- Different from other papers,by deriving the nonlinear performance functions of maintaining connectivity and avoiding collisions, the tracking performance of the formation in the transition and steady state can be adjusted, and the obstacle avoidance strategy is proposed under the premise of ensuring the tracking performance of the formation.

This manuscript is structured in the following manner: Section 2 delineates the problem statement. In Section 3, a formation tracking control design that incorporates connectivity-maintaining obstacle avoidance is developed, and an analysis of the stability of the proposed control system is conducted. The paper's conclusions are detailed in Section 4.

## II. PROBLEM STATEMENT

### A. Model of multiple nonholonomic mobile robots

Envision a scenario involving multiple nonholonomic mobile robots that are subject to both communication and sensing range limitations. Within the context of this study, the leader and follower roles are distinguished using the subscripts "i" and "j", respectively. The configuration of the follower, designated as j, is delineated in further detail below:

$$\dot{x}_j = \nu_j \cos\varphi_j - \omega_j a_j \sin\varphi_j$$
$$\dot{y}_j = \nu_j \sin\varphi_j + \omega_j a_j \cos\varphi_j \qquad (1)$$
$$\dot{\varphi}_j = \omega_j$$

$$H_j \dot{\eta}_j = -C_j(\eta_j)\eta_j - D_j\eta_j + \tau_{j,d} + B_j\tau_j^f \qquad (2)$$

In (1), $[x_j, y_j, \theta_j]^\top$ is the posture vector denoting the position $(x_j, y_j)$ and the orientation $\theta_j$ at the center of mass of the robot, $[\nu_j, \omega_j]^\top$; $\nu_j$ and $\omega_j$ denote the linear and angular velocity, respectively, $\tau_{j,d} = [\tau_{j,d1}, \tau_{j,d2}]^\top$ is the unknown disturbance, and $\tau_j^f = [\tau_{j,r}^f, \tau_{j,l}^f]^\mathrm{T}$ is the robot control torque of the follower j with an unknown fault. According to the literature [18], intermittent actuator failures can be expressed as:

$$\tau_{j,r}^f(t) = \rho_{j\nu k}(t)\tau_{j,r}(t) + \zeta_{j\nu k}(t), t \in [t_{j\nu k,s}, t_{j\nu k,e}) \qquad (3)$$

$$\tau_{j,l}^f(t) = \rho_{j\omega k}(t)\tau_{j,l}(t) + \zeta_{j\omega k}(t), t \in [t_{j\omega k,s}, t_{j\omega k,e}) \qquad (4)$$

where, $\rho_{j\nu k}(t) \in [0,1]$ , $\rho_{j\omega k}(t) \in [0,1]$ indicate the unknown failure rate of the follower j actuator, $k = 1, 2, 3, ...$ represents the k-th failure model. $t_{j\nu k,s}$ , $t_{j\omega k,s}$ and $t_{j\nu k,e}$ , $t_{j\omega k,e}$ are the time when the fault occurs and ends, respectively. In (2), the matrices are given as follows:

$$H_j = \begin{bmatrix} h_{j,1} & 0 \\ 0 & h_{j,2} \end{bmatrix}, C_j(\eta_j) = \begin{bmatrix} 0 & -b_j c_j \omega_j \\ \frac{c_j}{b_j}\omega_j & 0 \end{bmatrix}, D_j =$$

$$\frac{1}{2}\begin{bmatrix} d_{j,1} & d_{j,2} \\ d_{j,3} & d_{j,4} \end{bmatrix}, \quad B_j = \frac{r_j}{2b_j}\begin{bmatrix} b_j & b_j \\ 1 & -1 \end{bmatrix}, \text{with}$$

$$h_{j,1} = \frac{1}{2}r_j^2 m_j + I_{j,w}, h_{j,2} = \frac{1}{2}r_j^2 m_j + \frac{r_j^2}{2b_j^2}I_j + I_{j,w},$$

$$m_j = m_{j,c} + 2m_{j,w}, c_j = \frac{1}{2b_j}r_j^2 m_{j,c}a_j,$$

$$I_j = m_{j,c}a_j^2 + 2m_{j,w}b_j^2 + I_{j,c} + 2I_{j,m},$$

$$d_{j,1} = \frac{1}{2}\left(d_{j,11} + d_{j,22}\right), d_{j,2} = \frac{b_j}{2}\left(d_{j,11} - d_{j,22}\right),$$

$$d_{j,3} = \frac{1}{2b_j}\left(d_{j,11} - d_{j,22}\right), \quad d_{j,4} = \frac{1}{2}\left(d_{j,11} + d_{j,22}\right)$$

in this context, $H_j$ represents a symmetric, positive-definite inertia matrix. The variable $a_j$ specifies the distance from the follower $j$'s rear axle to its center of mass. The term $r_j$ is the radius of the driving wheels, while $b_j$ delineates the separation between the driving wheels and the axis of symmetry. The masses $m_{j,c}$ and $m_{j,w}$ correspond to the body and the wheel incorporating a motor, respectively. The moments of inertia are denoted as $I_{j,c}$ for the body around the vertical axis passing through its center of mass, $I_{j,w}$ for the wheel around its own axis, and $I_{j,m}$ for the wheel about its diameter. Additionally, $d_{j,11}$ and $d_{j,22}$ are the associated damping coefficients.In addition, the matrices $H_j$, $C_j$, $D_j$, and $B_j$ are unknown.The external interference $\tau_{j,d}$ is also unknown.

## B. Leader-follower model

We consider that the trajectory of Leader i satisfies the kinematics of a nonholonomic mobile robot, i.e., the trajectory is generated by the following kinetic equations:

$$\dot{x}_i = \nu_i \cos \varphi_i - \omega_i a_i \sin \varphi_i$$
$$\dot{y}_i = \nu_i \sin \varphi_i + \omega_i a_i \cos \varphi_i$$
$$\dot{\varphi}_i = \omega_i$$

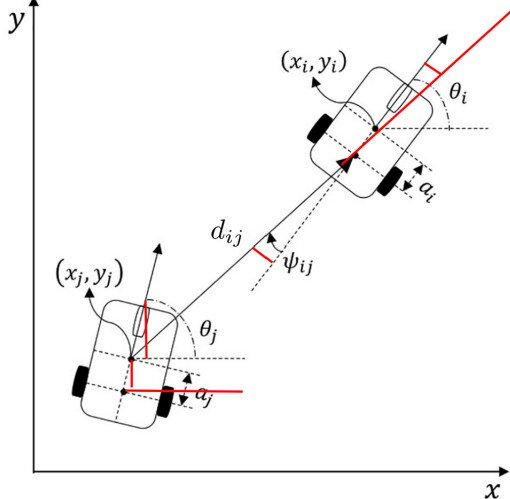

Fig. 1.  Leader-follower model

The leader-follower is described by the relative distance $d_{ij} = \sqrt{(\overline{x}_i - x_j)^2 + (\overline{y}_i - y_j)^2}$ and angle $\psi_{ij} = \theta_i - \arctan((\overline{y}_i - y_j)/(\overline{x}_i - x_j))$ between the leader i and the follower j where $\overline{x}_i = x_i - a_i\cos\theta_i$ and $\overline{y}_i = y_i - a_i\sin\theta_i$ (see Fig.1).The dynamics of the leader-follower model is obtained as

$$\dot{d}_{ij} = -\nu_j\cos\varphi_{ij} + a_j\omega_j\sin\varphi_{ij} + \nu_i\cos\psi_{ij}$$
$$\dot{\psi}_{ij} = \frac{1}{d_{ij}}\left(\nu_j\sin\varphi_{ij} + a_j\omega_j\cos\varphi_{ij} - \nu_i\sin\psi_{ij}\right) + \omega_i \tag{5}$$

where $\varphi_{ij} = \psi_{ij} - \theta_{ij}$ and $\theta_{ij} = \theta_i - \theta_j$. In Fig.1,In the process of information transmission of the mobile robot, the follower j can obtain the position and orientation of the leader i through its own sensor, or the directed graph can represent the communication topology between leader i and follower j. Moreover, follower j has the potential to become the leader for other nonholonomic mobile robots. In this case, a directed graph is also used to depict the communication between follower j and other nonholonomic mobile robots.As a result,the leader-follower paradigm can be seamlessly extended to encompass multiple autonomous mobile robots. In this configuration, the comprehensive communication network forms a directed spanning tree, with the leading robot 'i' serving as the foundational root node. This characteristic of the leader-follower approach highlights its significant scalability [17] (Fig.2).Therefore, this study focuses on the interplay between the lead robot, designated as i, and the follower robot,

labeled as j, in addition to the development of the formation control mechanism for the follower robot j.

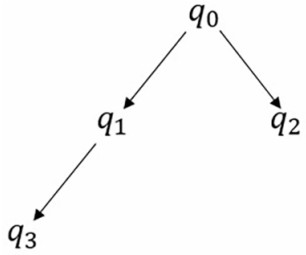

Fig. 2.  Initial connectivity of multiple mobile robots.

**Assumption 1.** The liner velocity $\nu_i$ and angular velocity $\omega_i$ of the leader i are bounded.

## C. Obstacle avoidance strategy

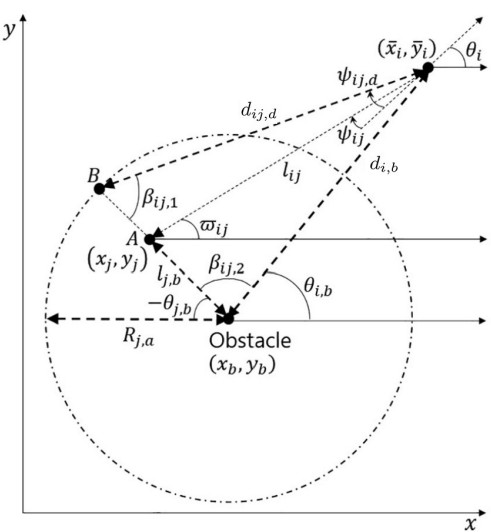

Fig. 3.  Obstacle avoidance strategy

In Fig.3, the angle $\overline{\psi}_{ij,d}$ is a predetermined value intended to implement an obstacle avoidance strategy that sustains connectivity. By applying the principles of trigonometry, specifically the cosine rule, $\overline{\psi}_{ij,d}$ can be expressed as follows:

$$\overline{\psi}_{ij,d} = \begin{cases} \psi_{ij,d} + \vartheta_{ij}\tanh\left(R_{j,a} - d_{j,b}\right), & d_{j,b} \leqslant R_{j,a} \\ \psi_{ij,d}, & d_{j,b} > R_{j,a} \end{cases} \tag{6}$$

where

$$\vartheta_{ij} = \varpi_i + \beta_{ij}\tanh\left(\frac{\varpi_i}{\gamma_{j,2}}\right) - \psi_{ij,d},$$
$$\varpi_i = \theta_i - \theta_{i,b},$$
$$\beta_{ij} = \beta_{ij,1} + \beta_{ij,2} - \pi,$$
$$\beta_{ij,1} = \arccos\left(\frac{d_{ij,d}^2 + R_{j,a}^2 - d_{i,b}^2}{2d_{ij,d}R_{j,a}}\right),$$
$$\beta_{ij,2} = \arccos\left(\frac{d_{i,b}^2 + R_{j,a}^2 - d_{ij,d}^2}{2d_{i,b}R_{j,a}}\right)$$

In this scenario, $R_{j,a}$ is defined as an avoidance initiation range that will be determined later, while $\gamma_{j,2}$ is a positive constant. The symbols $d_{j,b}$ and $d_{i,b}$ represent the distances from the follower $j$ and the leader $i$ to the obstacle, respectively. The variable $\theta_{i,b}$ stands for the relative angle of leader $i$ with respect to the obstacle. When $d_{i,b}$ exceeds $D_i$ and $d_{j,b}$ is less than or equal to $R_{j,a}$, where $D_i$ is the leader $i$'s sensing range, it implies that the follower $j$ might collide with the obstacle, as the follower cannot access the distance information $d_{i,b}$ described in equation (6). To avoid this issue, the values of $d_{i,b}$ and $\theta_{i,b}$ can be determined utilizing the relative data between follower $j$ and leader $i$ as follows:

$$
\begin{aligned}
d_{i,b} &= \sqrt{g_{ij,1}^2 + g_{ij,2}^2} \\
\theta_{i,b} &= \arctan(g_{ij,1}/g_{ij,2})
\end{aligned}
\tag{7}
$$

where $g_{ij,1} = d_{ij}\sin\varpi_{ij} - d_{j,b}\sin\theta_{j,b}$ , $g_{ij,2} = d_{ij}\cos\varpi_{ij} - d_{j,b}\cos\theta_{j,b}$ ,and $\theta_{j,b}$ is the relative angle of the obstacle from the follower j. Here, $\varpi_{ij}$ is measured using the local sensors of the follower j.

The current obstacle avoidance strategies [18] [19], which adjust the desired distance and angle, are not applicable for this paper.The reason for this is that the network of robots, which are constrained by their limited communication and sensory capabilities, will become disconnected if follower j navigates beyond the communication range of leader i during obstacle evasion. Therefore, we introduce a novel strategy for maintaining connectivity while avoiding obstacles (6), as depicted in Fig. 3.

### D. Formation control objective

The main purpose of this paper is to design the control matrices $\tau_{j,r}$ and $\tau_{j,l}$ so that the follower j can avoid collisions with obstacles while ensuring connectivity with the leader i, and at the same time, achieve control objectives,i.e.,

(1) Connectivity preservation and collision avoidance: $R_{j,m} < d_{ij}(t) < D_j, \forall t \geqslant 0$.

(2) Desired formation : $\lim_{t\to\infty} |d_{ij}(t) - d_{ij,d}| < \epsilon_{j,1}$, $\lim_{t\to\infty} |\psi_{ij}(t) - \psi_{ij,d}| < \epsilon_{j,2}$.

(3) Obstacle avoidance : $d_{i,b}(t) > R_{i,m}, \forall t \geqslant 0$.

In this context, $R_{j,m}$ symbolizes the minimal evasion radius for the follower designated as j. Concurrently, $D_j$ epitomizes the smallest extent of communication and sensory perception pertaining to the same follower j. Furthermore, the notations $d_{ij,d}$ and $\theta_{ij,d}$ are indicative of the optimal distance and angle that should exist between the leader i and the follower j to ensure the realization of the intended formation. Moreover, $\epsilon_{j,1}$ and $\epsilon_{j,2}$ represent pre-established positive constants. Lastly, $d_{j,b}$ signifies the measurement of the gap between the follower j and the impediment.

**Assumption 2.** The desired distance $d_{ij,d}$ and angle $\psi_{ij,d}$ satisfy $R_{j,m} < d_{ij,d} < D_j$ and $-\pi/2 \leqslant \psi_{ii,d} \leqslant \pi/2$, respectively.

## III. ADAPTIVE ASYMPTOTIC TRACKING FORMATION CONTROL DESIGN

In this section, we design a leader formation controller based on specified performance, and design an adaptive law to solve the problem of unknown failures in the actuator, which is able to avoid obstacles while maintaining connectivity, which is suitable for uncertain nonholonomic mobile robots with limited communication and sensing distance.

Let us define the errors as

$$
\begin{aligned}
e_{j,1} &= d_{ij} - d_{ij,d}, \\
e_{j,2} &= \psi_{ij} - \psi_{ij,f}, \\
e_{j,3} &= \theta_{j,a} - \theta_j, \\
e_{j,4} &= v_j - v_{j,\nu}, \\
e_{j,5} &= \omega_j - \omega_{j,\nu}
\end{aligned}
\tag{8}
$$

in the given paragraph, $\nu_{j,\nu}$ and $\omega_{j,\nu}$ represent the virtual control signals for follower j. The term $\theta_{j,a}$ is a virtual heading angle employed to address the kinematics' underactuated issue. Additionally, $\psi_{ij,f}$ signifies the filtered signal derived from the equation $\gamma_{j,1}\dot{\psi}_{ij,f} + \psi_{ij,f} = \overline{\psi}_{ij,d}$, where $\gamma_{j,1}$ is a small constant greater than zero. The desired relative angle, denoted as $\overline{\psi}_{ij,d}$, is illustrated in Fig. 3.

**Remark 1.** If $e_{j,2}$ is characterized as $e_{j,2} = \psi_{ij} - \psi_{ij,d}$, then the derivative of $e_{j,2}$ over time encompasses the velocity data of follower $j$, which is associated with the virtual control directives $v_{j,v}$ and $\omega_{j,v}$. This adds complexity to the design of the virtual controllers. To mitigate this complexity, $e_{j,2}$ in equation (8) is redefined as $e_{j,2} = \psi_{ij} - \psi_{ij,f}$, utilizing the signal $\psi_{ij,f}$ derived from the first-order low-pass filter $\gamma_{j,1}\dot{\psi}_{ij,f} + \psi_{ij,f} = \psi_{ij,d}$. Consequently, the derivative of $\psi_{ij,f}$ over time can be determined by $\dot{\psi}_{ij,f} = (\psi_{ij,d} - \psi_{ij,f})/\gamma_{j,1}$, excluding the velocity information of follower $j$.

**Remark 2.** Given the inherent underactuated characteristics of mobile robots, synchronizing the orientation of all robots becomes unachievable when navigating curved paths. To counteract this limitation, $e_{j,3}$ from equation (8) is redefined as $e_{j,3} = \theta_{j,a} - \theta_j$, where a new term, $\theta_{j,a}$, represents a virtual heading angle rather than $\theta_i$. This novel angle, $\theta_{j,a}$, will be formulated through a differential equation that serves as a virtual control law.

In the context of predefined performance-based leader-follower formation control, we introduce connectivity-maintaining and collision-avoiding performance functions to address the challenges of maintaining connectivity and avoiding collisions among robots, in line with the prescribed performance concept as described in reference [20].

The error surfaces, which are normalized through the connectivity-maintaining and collision-avoiding performance functions, are presented in (8).

$$
z_{j,1} = \frac{2e_{j,1} + \rho_{j,1}^L - \rho_{j,1}^U}{\rho_{j,1}^L + \rho_{j,1}^U}, \quad z_{j,n} = \frac{e_{j,n}}{\rho_{j,n}}, \quad n = 2,...,5 \tag{9}
$$

where

$$\rho_{j,1}^U(t) = \left(\rho_{j,1}^U(0) - \rho_{j,1}^U(\infty)\right)e^{-\lambda_{j,1}t} + \rho_{j,1}^U(\infty)$$
$$\rho_{j,1}^L(t) = \left(\rho_{j,1}^L(0) - \rho_{j,1}^L(\infty)\right)e^{-\lambda_{j,1}t} + \rho_{j,1}^L(\infty) \quad (10)$$
$$\rho_{j,n}(t) = (\rho_{j,n}(0) - \rho_{j,n}(\infty))e^{-\lambda_{j,n}t} + \rho_{j,n}(\infty)$$

with $\lambda_{j,1} > 0$ and $\lambda_{j,n} > 0$. In (10), the constants $\rho_{j,1}^U(0)$, $\rho_{j,1}^L(0)$, $\rho_{j,1}^U(\infty)$, $\rho_{j,1}^L(\infty)$, $\rho_{j,n}(0)$, and $\rho_{j,n}(\infty)$ are selected to satisfy the following conditions :

$$(i)\ 0 < \rho_{j,1}^U\left(\infty\right) < \rho_{j,1}^U\left(0\right) \leqslant D_j - d_{ij,d}$$
$$(ii)\ -\rho_{j,1}^L\left(0\right) < e_{j,1}\left(0\right) < \rho_{j,1}^U\left(0\right)$$
$$(iii)\ 0 < \rho_{j,1}^L\left(\infty\right) < \rho_{j,1}^L\left(0\right) \leqslant d_{ij,d} - R_{j,m} \quad (11)$$
$$(iv)\ \left|e_{j,2}(0)\right| < \rho_{j,2}(0) = \pi, \quad 0 < \rho_{j,2}(\infty) < \rho_{j,2}(0)$$
$$(v)\ \left|e_{j,p}(0)\right| < \rho_{j,p}(0), \quad 0 < \rho_{j,p}(\infty) < \rho_{j,p}(0)$$

where $p = 3, 4, 5$. It is important to note that the exponentially decaying performance functions in equation (8) and their time derivatives are inherently bounded. Consequently, the error surfaces $\varepsilon_{j,n}$ used for control design are defined as $\varepsilon_{j,n} = \ln((1 + z_{j,n})/(1 - z_{j,n}))$, following the prescribed performance design as outlined by [20], with $n = 1, \ldots, 5$.

**Remark 3.** The boundedness of $\varepsilon_{j,n}$ ensures that the absolute value of $z_{j,n}$ remains less than 1 for all times t greater than or equal to 0, with $n$ ranging from 1 to 5. When equation (9) meets the condition (11), the aforementioned inequality for $z_{j,n}$ implies that the error $e_{j,1}$ is confined between the negative lower limit of $\rho_{j,1}$ and the upper limit of $\rho_{j,1}$, while the magnitude of $e_{j,p}$ is maintained below $\rho_{j,p}$, $p = 1, \ldots, 5$. Consequently, by employing the predefined functions, the control objectives (1) and (2) can be successfully achieved. Therefore, our goal in formation tracking is to construct a tracker such that the boundedness of $\varepsilon_{j,n}$ is guaranteed through the principles of Lyapunov stability theory.

Design the virtual controller $v_{j,\nu}$, $\omega_{j,\nu}$, $\dot{\theta}_{j,a}$, $\alpha_{j\omega}$, $\alpha_{j\omega}$ as follows:

$$v_{j,\nu} = k_{j,1}\varepsilon_{j,1}\cos\varphi_{ij} - k_{j,2}\varepsilon_{j,2}\sin\varphi_{ij} \quad (12)$$

$$\omega_{j,\nu} = -\frac{k_{j,1}\varepsilon_{j,1}}{a_j}\sin\varphi_{ij} - \frac{k_{j,2}\varepsilon_{j,2}}{a_j}\cos\varphi_{ij} \quad (13)$$

$$\dot{\theta}_{j,a} = -k_{j,3}\varepsilon_{j,3} + \omega_{j,\nu} \quad (14)$$

$$\alpha_{j\nu} = \frac{1}{2}k_{j,4}\varepsilon_{j,4} + \frac{1}{2}k_{j,5}\varepsilon_{j,5} + \hat{\zeta}_{j\nu k}$$
$$+ \frac{\sigma_1\left(1 - z_{j,4}^2\right)\rho_{j,4}}{4\varepsilon_{j,4}}\hat{\rho}_{j\nu k} + \frac{\sigma_2\left(1 - z_{j,5}^2\right)\rho_{j,5}}{4\varepsilon_{j,5}}\hat{\rho}_{j\nu k}$$
$$(15)$$

$$\alpha_{j\omega} = \frac{1}{2}k_{j,4}\varepsilon_{j,4} - \frac{1}{2}k_{j,5}\varepsilon_{j,5} + \hat{\zeta}_{j\omega k}$$
$$+ \frac{\sigma_1\left(1 - z_{j,4}^2\right)\rho_{j,4}}{4\varepsilon_{j,4}}\hat{\rho}_{j\nu k} - \frac{\sigma_2\left(1 - z_{j,5}^2\right)\rho_{j,5}}{4\varepsilon_{j,5}}\hat{\rho}_{j\omega k}$$
$$(16)$$

where $k_{j,n} > 0$, $\sigma_1 > 0$, $\sigma_2 > 0$ are positive constants, $n = 1, \ldots, 5$.

The adaptive laws of the design parameters $\dot{\hat{\zeta}}_{j\nu k}$, $\dot{\hat{\zeta}}_{j\omega k}$, $\dot{\hat{\rho}}_{j\nu k}$, and $\dot{\hat{\rho}}_{j\omega k}$ are as follows:

$$\dot{\hat{\zeta}}_{j\nu k} = \frac{2\varepsilon_{j,4}}{\left(1 - z_{j,4}^2\right)\rho_{j,4}} + \frac{2\varepsilon_{j,5}}{\left(1 - z_{j,5}^2\right)\rho_{j,5}} - \lambda_1\hat{\zeta}_{j\nu k} \quad (17)$$

$$\dot{\hat{\zeta}}_{j\omega k} = \frac{2\varepsilon_{j,4}}{\left(1 - z_{j,4}^2\right)\rho_{j,4}} - \frac{2\varepsilon_{j,5}}{\left(1 - z_{j,5}^2\right)\rho_{j,5}} - \lambda_2\hat{\zeta}_{j\omega k} \quad (18)$$

$$\dot{\hat{\rho}}_{j\nu k} = \sigma_1 - \frac{1}{\hat{\rho}_{j\nu k}}\left[\frac{2\varepsilon_{j,4}}{\left(1 - z_{j,4}^2\right)\rho_{j,4}} + \frac{2\varepsilon_{j,5}}{\left(1 - z_{j,5}^2\right)\rho_{j,5}}\right]\alpha_{j\nu}$$
$$- \sigma_1\hat{\rho}_{j\nu k}$$
$$(19)$$

$$\dot{\hat{\rho}}_{j\omega k} = \sigma_2 - \frac{1}{\hat{\rho}_{j\omega k}}\left[\frac{2\varepsilon_{j,4}}{\left(1 - z_{j,4}^2\right)\rho_{j,4}} + \frac{2\varepsilon_{j,5}}{\left(1 - z_{j,5}^2\right)\rho_{j,5}}\right]\alpha_{j\omega}$$
$$- \sigma_2\hat{\rho}_{j\omega k}$$
$$(20)$$

where $\lambda_1 > 0$, $\lambda_2 > 0$ are positive constants, $\hat{\zeta}_{jmk} = \zeta_{jmk} - \tilde{\zeta}_{jmk}$ is an estimate of $\zeta_{jmk}$, $\hat{\rho}_{jmk} = \rho_{jmk} - \tilde{\rho}_{jmk}$ is an estimate of $\rho_{jmk}$, $k = \nu, \omega$.

Design controllers $\tau_{j,r}$ and $\tau_{j,l}$ as follows:

$$\tau_{j,r} = -\frac{\left[\frac{2\varepsilon_{j,4}}{(1-z_{j,4}^2)\rho_{j,4}} + \frac{2\varepsilon_{j,5}}{(1-z_{j,5}^2)\rho_{j,5}}\right]\hat{g}_1^2\alpha_{j\nu}^2}{\sqrt{\left[\frac{2\varepsilon_{j,4}}{(1-z_{j,4}^2)\rho_{j,4}} + \frac{2\varepsilon_{j,5}}{(1-z_{j,5}^2)\rho_{j,5}}\right]^2\hat{g}_1^2\alpha_{j\nu}^2 + \sigma_1^2}} \quad (21)$$

$$\tau_{j,l} = -\frac{\left[\frac{2\varepsilon_{j,4}}{(1-z_{j,4}^2)\rho_{j,4}} - \frac{2\varepsilon_{j,5}}{(1-z_{j,5}^2)\rho_{j,5}}\right]\hat{g}_2^2\alpha_{j\omega}^2}{\sqrt{\left[\frac{2\varepsilon_{j,4}}{(1-z_{j,4}^2)\rho_{j,4}} - \frac{2\varepsilon_{j,5}}{(1-z_{j,5}^2)\rho_{j,5}}\right]^2\hat{g}_2^2\alpha_{j\omega}^2 + \sigma_2^2}} \quad (22)$$

where $\sigma_1 > 0$, $\sigma_2 > 0$ are positive constants, $\hat{g}_1 = \frac{1}{\hat{\rho}_{j\nu k}}$, $\hat{g}_2 = \frac{1}{\hat{\rho}_{j\omega k}}$.

The primary finding of this study is encapsulated in the subsequent theorem: Examine nonholonomic mobile robotic systems represented by equations (1) and (2), which incorporate uncertainties. Given Assumptions 1 and 2, the formation regulations delineated in expressions (12)-(22), coupled with the connectivity-preserving obstacle circumvention strategy denoted by equation (6), where $R_{j,a} = d_{j,a} + R_{j,m}$ as referenced in [21], successfully fulfill the designated control objectives.,where

$$d_{j,a} = \sqrt{d_{ij,d}^2 + \left(d_{ij,d} + \rho_{j,1}^U\right)^2 - 2d_{ij,d}\left(d_{ij,d} - \rho_{j,1}^L\right)\cos(\rho_{j,2})}$$
$$+ \gamma_{j,b}$$
$$(23)$$

**Proof.** Substituting (1),(2), and (8), we have

$$\dot{z}_{j,1} = \frac{2}{\rho_{j,1}^L + \rho_{j,1}^U}\left(-\nu_{j,\nu}\cos\varphi_{ij} + a_j\omega_{j,\nu}\sin\varphi_{ij} + f_{j,1}\right) \quad (24)$$

$$\dot{z}_{j,2} = \frac{1}{\rho_{j,2}d_{ij}}\left(v_{j,\nu}\sin\varphi_{ij} + a_j\omega_{j,\nu}\cos\varphi_{ij} + f_{j,2}\right) \quad (25)$$

$$\dot{z}_{j,3} = \frac{1}{\rho_{j,3}}\left(\dot{\theta}_{j,a} - \omega_{j,\nu} + f_{j,3}\right) \quad (26)$$

$$\dot{z}_{j,4} = \frac{1}{\rho_{j,4}} \left( \sigma_{j,1} \left( \tau_{j,r}^f + \tau_{j,l}^f \right) + f_{j,4} \right) \tag{27}$$

$$\dot{z}_{j,5} = \frac{1}{\rho_{j,5}} \left( \sigma_{j,1} \left( \tau_{j,r}^f - \tau_{j,l}^f \right) + f_{j,5} \right) \tag{28}$$

where

$$f_{j,1} = \nu_i \cos\psi_{ij} - z_{j,4}\rho_{j,4}\cos\varphi_{ij} + a_j z_{j,5}\rho_{j,5}\sin\varphi_{ij}$$
$$+ \frac{1}{2} \left( \dot{\rho}_{j,1}^L - \dot{\rho}_{j,1}^U - z_{j,1} \left( \dot{\rho}_{j,1}^L + \dot{\rho}_{j,1}^U \right) \right)$$

$$f_{j,2} = z_{j,4}\rho_{j,4}\sin\varphi_{ij} + a_j z_{j,5}\rho_{j,5}\cos\varphi_{ij} - \nu_i\sin\psi_{ij}$$
$$+ \left( \frac{1}{2} \left( \left( \rho_{j,1}^L + \rho_{j,1}^U \right) z_{j,1} - \rho_{j,1}^L + \rho_{j,1}^U \right) + d_{ij,d} \right)$$
$$\times \left( \omega_i - \frac{\overline{\psi}_{ij,d} - \psi_{ij,f}}{\gamma_{j,1}} - z_{j,2}\dot{\rho}_{j,2} \right)$$

$$f_{j,3} = -z_{j,5}\rho_{j,5} - z_{j,3}\dot{\rho}_{j,3}$$

$$\begin{bmatrix} f_{j,4} \\ f_{j,5} \end{bmatrix} = H_j^{-1} \Big[ - \Big( C_j \big( \eta_j \big) + D_j \Big) \eta_j + \tau_{j,d} \Big]$$
$$- \begin{bmatrix} \dot{v}_{j,\nu}\dot{\omega}_{j,\nu} \end{bmatrix} - \begin{bmatrix} z_{j,4}\dot{\rho}_{j,4}z_{j,5}\dot{\rho}_{j,5} \end{bmatrix}$$

$$\eta_j = \begin{bmatrix} z_{j,4}\rho_{j,4} + \nu_{j,\nu} \\ z_{j,5}\rho_{j,5} + \omega_{j,\nu} \end{bmatrix}, \sigma_{j,1} = \frac{r_j}{2m_{j,1}}, \sigma_{j,2} = \frac{r_j}{2b_j m_{j,2}}.$$

Note that $\sigma_{j,1}$ and $\sigma_{j,2}$ are positive constants.

Let us define the open set $\Omega_z = \Omega_{z_1} \times \cdots \times \Omega_{z_5}$, where $\Omega_{z_n} \in (-1, 1)$ for $n = 1, \ldots, 5$. Given that the performance functions are chosen to meet the initial conditions (ii), (iv), and (v) specified in (11), we have $z_{j,n}(0) \in \Omega_{z_n}$. The control laws (12)–(22) are smooth over $\Omega_z$. Under Assumptions 1 to 2, and the external interference $\tau_{j,d}$ is also unknown. The nonlinear functions $f_{j,n}$ for $n = 1, \ldots, 5$ are continuously differentiable over $\Omega_z$. Consequently, $\dot{z}_{j,n}$ in equations (24)–(28) are continuously differentiable with respect to $t$ and are locally Lipschitz in $z_{j,n}$ over $\Omega_z$. Therefore, according to Theorem 3.1 in [22], there exist unique and maximal solutions $z_{j,n}(t)$ for all $t \in [0, \infty)$ to the differential equations (24)–(28). This implies that $z_{j,n}(t) \in (-1, 1)$ for all $t \in [0, \infty)$, and thus $\varepsilon_{j,n}$ are well-defined for all $t \in [0, \infty)$ where $n = 1, \ldots, 5$.

Then, the control laws (12)-(22) was demonstrated step by step ensure the boundedness of $\varepsilon_{j,n}$ for all $t \in [0, \infty)$.

step 1: Consider the following Lyapunov function

$$V_1(t) = \frac{1}{2}\varepsilon_{j,1}^2(t) + \frac{1}{2}\varepsilon_{j,2}^2(t) + \frac{1}{2}\varepsilon_{j,3}^2(t), t \in [0, \infty) \tag{29}$$

Using (12)-(14) and (24)-(26), the time derivative of (29) is

$$\dot{V}_1 = \frac{4\varepsilon_{j,1}}{\left(1 - z_{j,1}^2\right)\left(\rho_{j,1}^L + \rho_{j,1}^U\right)} \left(-k_{j,1}\varepsilon_{j,1} + f_{j,1}\right)$$
$$+ \frac{2\varepsilon_{j,2}}{\left(1 - z_{j,2}^2\right)d_{ij}\rho_{j,2}} \left(-k_{j,2}\varepsilon_{j,2} + f_{j,2}\right) \tag{30}$$
$$+ \frac{2\varepsilon_{j,3}}{\left(1 - z_{j,3}^2\right)\rho_{j,3}} \left(-k_{j,3}\varepsilon_{j,3} + f_{j,3}\right)$$

Due to the fact that $z_{j,n}(t)$ belongs to the interval (-1, 1) for all time instances within $[0, \infty)$, and considering Assumptions

1 and 2, as well as the constrained nature of the performance functions and their respective time derivatives—namely $f_{j,1}$, $f_{j,2}$, and $f_{j,3}$—it is evident that these functions remain bounded across the domain $\Omega_z$. Consequently, there exist unspecified positive constants, denoted as $\overline{f}_{j,1}$, $\overline{f}_{j,2}$, and $\overline{f}_{j,3}$, which ensure that

$$|f_{j,1}(t)| \leqslant \overline{f}_{j,1}, |f_{j,2}(t)| \leqslant \overline{f}_{j,2}, |f_{j,3}(t)| \leqslant \overline{f}_{j,3}. \tag{31}$$

According to (31), (30) can be rewritten as

$$\dot{V}_1 \leqslant \frac{4}{\left(1 - z_{j,1}^2\right)\left(\rho_{j,1}^L + \rho_{j,1}^U\right)} \left(-k_{j,1}|\varepsilon_{j,1}|^2 + \overline{f}_{j,1}|\varepsilon_{j,1}|\right)$$
$$+ \frac{2}{\left(1 - z_{j,2}^2\right)d_{ij}\rho_{j,2}} \left(-k_{j,2}|\varepsilon_{j,2}|^2 + \overline{f}_{j,2}|\varepsilon_{j,2}|\right)$$
$$+ \frac{2}{\left(1 - z_{j,3}^2\right)\rho_{j,3}} \left(- k_{j,3}\left|\varepsilon_{j,3}\right|^2 + \overline{f}_{j,3}\left|\varepsilon_{j,3}\right|\right)$$
$$\tag{32}$$

Owing to $z_{j,n}(t) \in (-1, 1)$ with $n = 1, 2, 3$, $\forall t \in [0, \infty)$, it holds that $4/((1 - z_{j,1}^2)(\rho_{j,1}^L + \rho_{j,1}^U)) > 0$, $2/((1 - z_{j,2}^2)d_{ij}\rho_{j,2}) > 0$, and $2/((1 - z_{j,3}^2)\rho_{j,3}) > 0$. Thus, $\varepsilon_{j,1}(t)$, $\varepsilon_{j,2}(t)$ and $\varepsilon_{j,3}(t)$ are bounded for all $t \in [0, \infty)$ such that

$$|\varepsilon_{j,1}(t)| \leqslant \overline{\varepsilon}_{j,1} = \max \left\{ |\varepsilon_{j,1}(0)|, \frac{\overline{f}_{j,1}}{k_{j,1}} \right\}$$
$$|\varepsilon_{j,2}(t)| \leqslant \overline{\varepsilon}_{j,2} = \max \left\{ |\varepsilon_{j,2}(0)|, \frac{\overline{f}_{j,2}}{k_{j,2}} \right\} \tag{33}$$
$$|\varepsilon_{j,3}(t)| \leqslant \overline{\varepsilon}_{j,3} = \max \left\{ |\varepsilon_{j,3}(0)|, \frac{\overline{f}_{j,3}}{k_{j,3}} \right\}.$$

From (12)-(14) and (33), it can deduce the boundedness of $\nu_{j,\nu}(t)$, $\omega_{j,\nu}(t)$, $\dot{\nu}_{j,\nu}(t)$ and $\dot{\omega}_{j,\nu}(t)$ for all $t \in [0, \infty)$.

step 2:Consider the following Lyapunov function for all $t \in [0, \infty)$.

$$V_2(t) = \frac{1}{2} \left( \frac{1}{\sigma_{j,1}}\varepsilon_{j,4}^2(t) + \frac{1}{\sigma_{j,2}}\varepsilon_{j,5}^2(t) \right)$$
$$+ \frac{1}{2}\tilde{\rho}_{j\nu k}^2 + \frac{1}{2}\tilde{\rho}_{j\omega k}^2 + \frac{1}{2}\tilde{\zeta}_{j\nu k}^2 + \frac{1}{2}\tilde{\zeta}_{j\omega k}^2 \tag{34}$$

The time derivation of (34) is

$$\dot{V}_2 = \frac{2\varepsilon_{j,4}}{\left(1 - z_{j,4}^2\right)\sigma_{j,4}}\dot{z}_{j,4} + \frac{2\varepsilon_{j,5}}{\left(1 - z_{j,5}^2\right)\sigma_{j,5}}\dot{z}_{j,5}$$
$$+ \tilde{\rho}_{j\nu k}\dot{\tilde{\rho}}_{j\nu k} + \tilde{\rho}_{j\omega k}\dot{\tilde{\rho}}_{j\omega k} + \tilde{\zeta}_{j\nu k}\dot{\tilde{\zeta}}_{j\nu k} + \tilde{\zeta}_{j\omega k}\dot{\tilde{\zeta}}_{j\omega k} \tag{35}$$

Substituting (27) and (28) into (35) yields

$$\dot{V}_2 = \frac{2\varepsilon_{j,4}}{\left(1 - z_{j,4}^2\right)\rho_{j,4}}\Big[\Big(\tau_{j,r}^f + \tau_{j,l}^f\Big) + \frac{f_{j,4}}{\sigma_{j,1}}\Big]$$
$$+ \frac{2\varepsilon_{j,5}}{\left(1 - z_{j,5}^2\right)\rho_{j,5}}\Big[\Big(\tau_{j,r}^f - \tau_{j,l}^f\Big) + \frac{f_{j,5}}{\sigma_{j,2}}\Big] \tag{36}$$
$$+ \tilde{\rho}_{j\nu k}\dot{\tilde{\rho}}_{j\nu k} + \tilde{\rho}_{j\omega k}\dot{\tilde{\rho}}_{j\omega k} + \tilde{\zeta}_{j\nu k}\dot{\tilde{\zeta}}_{j\nu k} + \tilde{\zeta}_{j\omega k}\dot{\tilde{\zeta}}_{j\omega k}$$

Substituting (3) and (4) into (36) yields

$$\dot{V}_2 = \frac{2\varepsilon_{j,4}}{\left(1 - z_{j,4}^2\right)\rho_{j,4}}[(\rho_{j\nu k}(t)\tau_{j,r}(t) + \zeta_{j\nu k}(t))$$
$$+ (\rho_{j\omega k}(t)\tau_{j,l}(t) + \zeta_{j\omega k}(t)) + (\alpha_{j\nu} + \alpha_{j\omega})$$
$$- (\alpha_{j\nu} + \alpha_{j\omega}) + \frac{f_{j,4}}{\sigma_{j,1}}]$$
$$+ \frac{2\varepsilon_{j,5}}{\left(1 - z_{j,5}^2\right)\rho_{j,5}}[(\rho_{j\nu k}(t)\tau_{j,r}(t) + \zeta_{j\nu k}(t)) \quad (37)$$
$$- (\rho_{j\omega k}(t)\tau_{j,l}(t) + \zeta_{j\omega k}(t)) + (\alpha_{j\nu} - \alpha_{j\omega})$$
$$- (\alpha_{j\nu} - \alpha_{j\omega}) + \frac{f_{j,5}}{\sigma_{j,2}}]$$
$$+ \tilde{\rho}_{j\nu k}\dot{\hat{\rho}}_{i\nu k} + \tilde{\rho}_{j\omega k}\dot{\hat{\rho}}_{j\omega k} + \tilde{\zeta}_{j\nu k}\dot{\hat{\zeta}}_{j\nu k} + \tilde{\zeta}_{j\omega k}\dot{\hat{\zeta}}_{j\omega k}$$

According to (21) and (22), there are the following inequalities

$$[\frac{2\varepsilon_{j,4}}{\left(1 - z_{j,4}^2\right)\rho_{j,4}} + \frac{2\varepsilon_{j,5}}{\left(1 - z_{j,5}^2\right)\rho_{j,5}}]\rho_{j\nu k}\tau_{j,r}$$
$$\leq -[\frac{2\varepsilon_{j,4}}{\left(1 - z_{j,4}^2\right)\rho_{j,4}} + \frac{2\varepsilon_{j,5}}{\left(1 - z_{j,5}^2\right)\rho_{j,5}}]\rho_{j\nu k}$$
$$\frac{[\frac{2\varepsilon_{j,4}}{(1-z_{j,4}^2)\rho_{j,4}} + \frac{2\varepsilon_{j,5}}{(1-z_{j,5}^2)\rho_{j,5}}]\hat{g}_1^2\alpha_{j\nu}^2}{\sqrt{[\frac{2\varepsilon_{j,4}}{(1-z_{j,4}^2)\rho_{j,4}} + \frac{2\varepsilon_{j,5}}{(1-z_{j,5}^2)\rho_{j,5}}]^2\hat{g}_1^2\alpha_{j\nu}^2 + \sigma_1^2}}$$
$$\leq -\rho_{j\nu k}\frac{[\frac{2\varepsilon_{j,4}}{(1-z_{j,4}^2)\rho_{j,4}} + \frac{2\varepsilon_{j,5}}{(1-z_{j,5}^2)\rho_{j,5}}]^2\hat{g}_1^2\alpha_{j\nu}^2}{\sqrt{[\frac{2\varepsilon_{j,4}}{(1-z_{j,4}^2)\rho_{j,4}} + \frac{2\varepsilon_{j,5}}{(1-z_{j,5}^2)\rho_{j,5}}]^2\hat{g}_1^2\alpha_{j\nu}^2 + \sigma_1^2}}$$
$$\leq \sigma_1\rho_{j\nu k} - \rho_{j\nu k}[\frac{2\varepsilon_{j,4}}{\left(1 - z_{j,4}^2\right)\rho_{j,4}} + \frac{2\varepsilon_{j,5}}{\left(1 - z_{j,5}^2\right)\rho_{j,5}}]\hat{g}_1\alpha_{j\nu}$$
$$(38)$$

$$[\frac{2\varepsilon_{j,4}}{\left(1 - z_{j,4}^2\right)\rho_{j,4}} - \frac{2\varepsilon_{j,5}}{\left(1 - z_{j,5}^2\right)\rho_{j,5}}]\rho_{j\omega k}\tau_{j,l}$$
$$\leq -[\frac{2\varepsilon_{j,4}}{\left(1 - z_{j,4}^2\right)\rho_{j,4}} - \frac{2\varepsilon_{j,5}}{\left(1 - z_{j,5}^2\right)\rho_{j,5}}]\rho_{j\omega k}$$
$$\frac{[\frac{2\varepsilon_{j,4}}{(1-z_{j,4}^2)\rho_{j,4}} - \frac{2\varepsilon_{j,5}}{(1-z_{j,5}^2)\rho_{j,5}}]\hat{g}_2^2\alpha_{j\nu}^2}{\sqrt{[\frac{2\varepsilon_{j,4}}{(1-z_{j,4}^2)\rho_{j,4}} - \frac{2\varepsilon_{j,5}}{(1-z_{j,5}^2)\rho_{j,5}}]^2\hat{g}_2^2\alpha_{j\nu}^2 + \sigma_1^2}}$$
$$\leq -\rho_{j\omega k}\frac{[\frac{2\varepsilon_{j,4}}{(1-z_{j,4}^2)\rho_{j,4}} - \frac{2\varepsilon_{j,5}}{(1-z_{j,5}^2)\rho_{j,5}}]^2\hat{g}_2^2\alpha_{j\omega}^2}{\sqrt{[\frac{2\varepsilon_{j,4}}{(1-z_{j,4}^2)\rho_{j,4}} - \frac{2\varepsilon_{j,5}}{(1-z_{j,5}^2)\rho_{j,5}}]^2\hat{g}_2^2\alpha_{j\omega}^2 + \sigma_1^2}}$$
$$\leq \sigma_2\rho_{j\omega k} - \rho_{j\omega k}[\frac{2\varepsilon_{j,4}}{\left(1 - z_{j,4}^2\right)\rho_{j,4}} - \frac{2\varepsilon_{j,5}}{\left(1 - z_{j,5}^2\right)\rho_{j,5}}]\hat{g}_2\alpha_{j\omega}$$
$$(39)$$

where $\sigma_1 > 0$, $\sigma_2 > 0$ are positive constants, $\hat{g}_1 = \frac{1}{\bar{\rho}_{j\nu k}}$, $\hat{g}_2 = \frac{1}{\bar{\rho}_{j\omega k}}$.

Substituting (15)-(20) and (38)-(39) into (37) yields

$$\dot{V}_2 \leq -\frac{2k_{j,4}}{\rho_{j,4}\left(1 - z_{j,4}^2\right)}\varepsilon_{j,4}^2 - \frac{2k_{j,5}}{\rho_{j,5}\left(1 - z_{j,5}^2\right)}\varepsilon_{j,5}^2$$
$$+ \lambda_1\tilde{\zeta}_{j\nu k}\hat{\zeta}_{j\nu k} + \lambda_2\tilde{\zeta}_{j\omega k}\hat{\zeta}_{j\omega k} + \sigma_1\tilde{\rho}_{j\nu k}\hat{\rho}_{j\nu k}$$
$$+ \sigma_2\tilde{\rho}_{j\omega k}\hat{\rho}_{j\omega k} + \frac{2k_{j,4}}{\rho_{j,4}\left(1 - z_{j,4}^2\right)}\frac{f_{j,4}}{\sigma_{j,1}} \quad (40)$$
$$+ \frac{2k_{j,5}}{\rho_{j,5}\left(1 - z_{j,5}^2\right)}\frac{f_{j,5}}{\sigma_{j,2}}$$

The following inequalities hold

$$\lambda_1\tilde{\zeta}_{j\nu k}\hat{\zeta}_{j\nu k} \leq -\frac{\lambda_1}{2}\tilde{\zeta}_{j\nu k}^2 + \frac{\lambda_1}{2}\zeta_{j\nu k}^2 \quad (41)$$

$$\lambda_2\tilde{\zeta}_{j\omega k}\hat{\zeta}_{j\omega k} \leq -\frac{\lambda_2}{2}\tilde{\zeta}_{j\omega k}^2 + \frac{\lambda_2}{2}\zeta_{j\omega k}^2 \quad (42)$$

$$\sigma_1\tilde{\rho}_{j\nu k}\hat{\rho}_{j\nu k} \leq -\frac{\sigma_1}{2}\tilde{\rho}_{j\nu k}^2 + \frac{\sigma_1}{2}\rho_{j\nu k}^2 \quad (43)$$

$$\sigma_2\tilde{\rho}_{j\omega k}\hat{\rho}_{j\omega k} \leq -\frac{\sigma_2}{2}\tilde{\rho}_{j\omega k}^2 + \frac{\sigma_2}{2}\rho_{j\omega k}^2 \quad (44)$$

Substituting (41)-(44) into (40) yields

$$\dot{V}_2 \leq -\frac{2k_{j,4}}{\rho_{j,4}\left(1 - z_{j,4}^2\right)}\varepsilon_{j,4}^2 - \frac{2k_{j,5}}{\rho_{j,5}\left(1 - z_{j,5}^2\right)}\varepsilon_{j,5}^2$$
$$- \frac{\lambda_1}{2}\tilde{\zeta}_{j\nu k}^2 - \frac{\lambda_2}{2}\tilde{\zeta}_{j\omega k}^2 + \frac{\lambda_1}{2}\zeta_{j\nu k}^2 + \frac{\lambda_2}{2}\zeta_{j\omega k}^2$$
$$- \frac{\sigma_1}{2}\tilde{\rho}_{j\nu k}^2 - \frac{\sigma_2}{2}\tilde{\rho}_{j\omega k}^2 + \frac{\sigma_1}{2}\rho_{j\nu k}^2 + \frac{\sigma_2}{2}\rho_{j\omega k}^2 \quad (45)$$
$$+ \frac{2k_{j,4}}{\rho_{j,4}\left(1 - z_{j,4}^2\right)}\frac{f_{j,4}}{\sigma_{j,1}} + \frac{2k_{j,5}}{\rho_{j,5}\left(1 - z_{j,5}^2\right)}\frac{f_{j,5}}{\sigma_{j,2}}$$

Owing to $z_{j,n}(t) \in (-1, 1)$, $\forall t \in [0, \infty)$, and the boundedness of $\nu_{j,\nu}(t)$, $\omega_{j,\nu}(t)$, $\dot{\nu}_{j,\nu}(t)$ and $\dot{\omega}_{j,\nu}(t)$ for all $t \in [0, \infty)$. $f_{j,5}$ and $f_{j,4}$, are bounded over $\Omega_z$. Thus, there are unknown constants $\overline{f}_{j,4} > 0$ and $\overline{f}_{j,5} > 0$ for all $t \in [0, \infty)$ satisfying

$$|f_{j,4}(t)/\sigma_{j,1}| \leqslant \overline{f}_{j,4},$$
$$|f_{j,5}(t)/\sigma_{j,2}| \leqslant \overline{f}_{j,5}, \quad \forall t \in [0, \infty). \quad (46)$$

Substituting (46) into (45) yields

$$\dot{V}_2 \leq -\frac{2k_{j,4}}{\rho_{j,4}\left(1 - z_{j,4}^2\right)}\varepsilon_{j,4}^2 - \frac{2k_{j,5}}{\rho_{j,5}\left(1 - z_{j,5}^2\right)}\varepsilon_{j,5}^2$$
$$- \frac{\lambda_1}{2}\tilde{\zeta}_{j\nu k}^2 - \frac{\lambda_2}{2}\tilde{\zeta}_{j\omega k}^2 + \frac{\lambda_1}{2}\zeta_{j\nu k}^2 + \frac{\lambda_2}{2}\zeta_{j\omega k}^2$$
$$- \frac{\sigma_1}{2}\tilde{\rho}_{j\nu k}^2 - \frac{\sigma_2}{2}\tilde{\rho}_{j\omega k}^2 + \frac{\sigma_1}{2}\rho_{j\nu k}^2 + \frac{\sigma_2}{2}\rho_{j\omega k}^2 \quad (47)$$
$$+ \frac{2k_{j,4}}{\rho_{j,4}\left(1 - z_{j,4}^2\right)}\overline{f}_{j,4} + \frac{2k_{j,5}}{\rho_{j,5}\left(1 - z_{j,5}^2\right)}\overline{f}_{j,5}$$

According to the definition of $V_2$ and (47), the final form of $\dot{V}_2$ is

$$\dot{V}_2 \leq -C_jV_2 + D_j \quad (48)$$

where,

$$C_j = \min\{\frac{2k_{j,4}}{\rho_{j,4}\left(1 - z_{j,4}^2\right)}, \frac{2k_{j,4}}{\rho_{j,4}\left(1 - z_{j,4}^2\right)}, \frac{\lambda_1}{2}, \frac{\lambda_2}{2}, \frac{\sigma_1}{2}, \frac{\sigma_2}{2}\}$$

$$D_j = \frac{\lambda_1}{2}\zeta_{j\nu k}^2 + \frac{\lambda_2}{2}\zeta_{j\omega k}^2 + \frac{\sigma_1}{2}\rho_{j\nu k}^2 + \frac{\sigma_2}{2}\rho_{j\omega k}^2$$
$$+ \frac{2k_{j,4}}{\rho_{j,4}\left(1-z_{j,4}^2\right)}\overline{f}_{j,4} + \frac{2k_{j,5}}{\rho_{j,5}\left(1-z_{j,5}^2\right)}\overline{f}_{j,5}$$

According to (48), [23] can be concluded that the controlled multi-incomplete mobile robot system is stable and all signals are bounded. The boundedness of $\varepsilon_{j,n}$, $n=1,\ldots,5$ gives

$$-1 < \frac{e^{-\overline{\varepsilon}_{j,n}}-1}{e^{-\overline{\varepsilon}_{j,n}}+1} = \underline{z}_{j,n} \leqslant z_{j,n}\left(t\right) \leqslant \overline{z}_{j,n} = \frac{e^{\overline{\varepsilon}_{j,n}}-1}{e^{\overline{\varepsilon}_{j,n}}+1} < 1 \tag{49}$$

From the boundedness of $\varepsilon_{j,1}$ and (9), we obtain

$$-\rho_{j,1}^L\left(t\right) < e_{j,1}\left(t\right) < \rho_{j,1}^U\left(t\right), \quad \forall t \geqslant 0. \tag{50}$$

By applying the conditions (i) and (iii) in (11),it holds that

$$R_{j,m} < d_{ij}\left(t\right) < D_j, \quad \forall t \geqslant 0 \tag{51}$$

Hence, the formation control objective (1) is ensured for all $t \geqslant 0$. Furthermore, in the absence of any collisions with obstacles, applying equation (49) results in:

$$\begin{aligned} \lim_{t\to\infty}\left|d_{ij}\left(t\right)-d_{ij,d}\right| &< \epsilon_{j,1} \\ \lim_{t\to\infty}\left|\psi_{ij}\left(t\right)-\psi_{ij,d}\right| &< \epsilon_{j,2} \end{aligned} \tag{52}$$

where $\epsilon_{j,1} = \max\{\rho_{j,1}^L(\infty), \rho_{j,1}^U(\infty)\}$ and $\epsilon_{j,2} = \max\{\rho_{j,2}(\infty), \rho_{j,2}(\infty)\}$. Hence, the formation control objective (2) is achieved.

Since $\overline{\psi}_{ij,d} = \psi_{ij,d}$ when $d_{j,b} > R_{j,a}$ and $\left|\psi_{ij,d}-\psi_{ij}\right| < \rho_{j,2}(0) = \pi$ ,it holds form Fig.3 that

$$\begin{aligned} d_{j,b} &= R_{j,a} - \sqrt{d_{ij,d}^2 + d_{ij}^2 - 2d_{ij,d}d_{ij}\cos(\psi_{ij,d}-\psi_{ij})} \\ &> R_{j,a} - d_{j,a} \end{aligned} \tag{53}$$

Substituting $R_{j,a} = d_{j,a} + R_{j,m}$ into (53), we have $d_{j,b} > R_{j,m}$. Therefore, the formation control objective (3) is guaranteed for all $t \geqslant 0$.

**Remark 4.** As shown in the reference [17], the stability of formation for a group consisting of $N+1$ robots, where there is one leader labeled as i and $N$ followers, can be validated through the aggregate sum of Lyapunov functions for each individual j. This is expressed as $V = \sum_{j=1}^{N} V_1 + V_2$.

## IV. CONCLUSIONS

Aiming at the uncertain nonholonomic mobile robot with limited communication and perception, the connectivity and obstacle avoidance problem of master-slave formation tracking based on specified performance under the presence of unknown actuator failure is studied. Compared to relevant literature, this paper's main contributions include: (1) deriving nonlinear error surfaces using connectivity-maintaining and collision-avoiding performance functions for prescribed-performance-based formation control tracking, without relying on potential-like functions; and (2) proposing an obstacle avoidance strategy that maintains connectivity between the leader and follower while avoiding obstacles in cases of unknown actuator failure. The corresponding controller and parameter adaptive law are designed.The theoretical analysis has been presented for the Lyapunov stability.

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
