# OpenReview forum: "Connectivity and Obstacle Avoidance Method for Formation Tracking with Uncertain Multiple Nonholonomic Mobile Robots with Unknown Faults"
_IEEE.org/ICIST/2024/Conference — IEEE ICIST 2024 Conference Submission_

### Official Review · Reviewer_Tewf · 2024-08-21
**Connectivity and Obstacle Avoidance Method for Formation Tracking with Uncertain Multiple Nonholonomic Mobile Robots with Unknown Faults**

**Rating:** 8
**Confidence:** 5

**Review:**

This paper delves into the challenge of ensuring connectivity and steering clear of obstacles for multiple uncertain nonholonomic mobile robots that face limitations in communication and sensing distances under the influence of unknown actuator defects. The authors provide the specific design process of the controller and stability analysis of the system. The design method used appears to be correct. However, the rationality of all the assumptions should be discussed. Furthermore, it is advisable to include simulation results to validate the effectiveness of the proposed method.

---

### Official Review · Reviewer_oTSV · 2024-08-27
**The topic under consideration is interesting. This paper can be accepted after minor modifications.**

**Rating:** 8
**Confidence:** 3

**Review:**

This paper investigates a connectivity and obstacle avoidance method for formation tracking with uncertain multiple nonholonomic mobile robots with unknown faults. The topic under consideration is interesting. Detailed comments and suggestions are listed as follows.
1.	The English writing of the paper needs to be further polished, and some typos should be fixed, such as “deriving nonlinear error surfaces using connectivity-maintaining and collision-avoiding performance functions for prescribed performance-based formation control tracking, without relying on potential-like functions”.
2.	On Page 4, the derivation of equation (9) cannot be easily followed. Please give a detailed explanation on the content.
3.	The format of References should be standardized.

---

### Decision · Program_Chairs · 2024-09-06

Accept (Oral)